# Establishing Human and Canine Xenograft Murine Osteosarcoma Models for Application of Focused Ultrasound Ablation

**DOI:** 10.3390/biomedicines13092122

**Published:** 2025-08-30

**Authors:** Alayna N. Hay, Alex Simon, Lauren N. Ruger, Jessica Gannon, Sheryl Coutermarsh-Ott, Elliana R. Vickers, William Eward, Nathan J. Neufeld, Eli Vlaisavljevich, Joanne Tuohy

**Affiliations:** 1Department of Small Animal Clinical Sciences, Virginia Maryland College of Veterinary Medicine, Blacksburg, VA 24061, USA; anw5137@vt.edu; 2Virginia Tech Animal Cancer Care and Research Center, Virginia Maryland College of Veterinary Medicine, Roanoke, VA 24016, USA; 3Department of Biomedical Engineering and Mechanics, Virginia Polytechnic Institute and State University, Blacksburg, VA 24061, USAlarnold24@vt.edu (L.N.R.); jess98@vt.edu (J.G.); eliv@vt.edu (E.V.); 4Department of Biomedical Sciences and Pathobiology, Virginia Maryland College of Veterinary Medicine, Blacksburg, VA 24060, USA; slc2003@vt.edu; 5Graduate Program in Translation Biology, Medicine and Health, Virginia Polytechnic Institute and State University, Roanoke, VA 24016, USA; ellianarv@vt.edu; 6Duke Cancer Institute, Duke University, Durham, NC 27710, USA; william.eward@duke.edu; 7Department of Orthopedic Surgery, Duke University, Durham, NC 27710, USA; 8Division of Pediatric Hematology/Oncology, Duke University Medical Center, Durham, NC 27710, USA; 9Department of Supportive Care, City of Hope, Atlanta, GA 30329, USA; 10Edward Via College of Osteopathic Medicine, Auburn University, Auburn, AL 36849, USA

**Keywords:** animal models for cancer, precision oncology, bone tumors, histotripsy

## Abstract

**Background:** Osteosarcoma (OS) is the most commonly occurring type of bone cancer in both humans and canines. The survival outcomes for OS patients have not improved significantly in decades. A novel and innovative treatment option that is currently under investigation for OS in the veterinary field is the focused ultrasound ablation modality, histotripsy. Histotripsy is a non-thermal, non-invasive, non-ionizing ablation modality that destroys tissue through generation of acoustic cavitation. **Objective:** In the current study, we sought to investigate the utility of an orthotropic OS xenograft murine model for characterization of chronic ablative and clinical outcomes post-histotripsy ablation. **Method:** Given the high comparative relevance of canine to human OS, histotripsy was delivered to orthotopic OS tumors in both human and canine xenograft murine models. **Results:** Histotripsy improved limb function in tumor-bearing mice compared to untreated tumor bearing mice. The results of this study demonstrated the utility of the orthotopic OS xenograft murine model for histotripsy-based preclinical studies. **Conclusions:** The current study is the first published investigation for the use of an orthotopic xenograft murine model for the development of histotripsy ablation for OS. The developmental process of the model, technical limitations, and future directions are discussed.

## 1. Introduction

Osteosarcoma (OS) is the most prevalent primary malignant bone tumor in humans, primarily affecting children, adolescents, and, to a lesser extent, individuals > 60 years old [1,2]. In humans, OS accounts for greater than 55% of primary bone tumors [3]. Definitive standard of care includes limb salvage or amputation surgery, along with neoadjuvant and adjuvant chemotherapy [2,4]. The median 5-year survival rate is 70%, with metastatic disease being the predominant cause of mortality [4,5].Unfortunately, the median survival for OS patients has not improved significantly since the addition of chemotherapy to standard of care treatment protocols decades ago, demonstrating the profound need to continue to advance OS treatment options and improve patient prognosis [4]. In addition to humans, the pet dog also develops spontaneously occurring OS, and the disease shares numerous genetic and biological similarities with human OS [1,6]. Canine OS occurs at a greater prevalence than human OS, and the shorter lifespan of the dog results in a compressed disease course relative to the human, allowing for more rapid completion of bench to bedside investigations [1,7]. These factors make the canine a valuable and informative comparative disease model.

To advance treatment options for both species and allow for further comparative investigations between human and canine OS, the preclinical murine xenograft model can be utilized [8,9,10,11]. This model allows for comprehensive and mechanistic studies to evaluate oncological outcomes associated with novel treatment options, such as vaccines [8] and drugs [9]. These evaluations cannot feasibly and ethically be conducted in human and veterinary patients. Furthermore, the murine xenograft model recapitulates OS tumor heterogeneity for characterization of patient-specific treatment responses [10,12,13].

Histotripsy is a non-thermal, non-invasive, non-ionizing ablation modality that destroys tissue through generation of acoustic cavitation [14]. Acoustic cavitation is the generation of microbubbles (cavitation bubble cloud) that rapidly expand and collapse, imparting high strain on surrounding cells and effectively disintegrating targeted tissue throughout the treatment delivery [14,15]. Unlike thermal ablation modalities, histotripsy is not subject to the heat sink effect, and the tissue-selective properties of histotripsy increase ablation precision and reduce the risk of off-target ablation and consequent tissue damage [15]. Our team has demonstrated the safety and feasibility of ablating canine bone tumors both ex vivo and short-term in vivo with histotripsy to allow for comparative oncology investigations [16,17,18]. However, for investigations aimed at studying mechanistic and comprehensive ablative and oncological outcomes post-histotripsy, a relevant pre-clinical rodent model would be beneficial. Previous reports have demonstrated the utility of heterotopic xenograft murine models for histotripsy-based investigations for neuroblastoma [19] and liver cancer [20,21]. Given the utility of the xenograft murine model for both OS and histotripsy investigations, the objective of the current study was to evaluate the functional and chronic ablative outcomes of the novel tumor ablation modality histotripsy for human and canine OS by utilizing a preclinical orthotopic xenograft murine model.

For the current study, we hypothesized that targeted regions of tumors would be successfully ablated with histotripsy using our previously developed 1 MHz histotripsy system, [22,23] which would be evident based on non-invasive imaging modalities. We further hypothesized that histotripsy ablation would improve limb function in the tumor-bearing limb compared to untreated tumor-bearing controls, demonstrating the potential of histotripsy to serve as a clinically relevant limb salvage treatment option for OS patients. To test these hypotheses, we utilized an immunocompromised mouse strain and experimentally induced human or canine bone tumors with commercially available OS cell lines to create both human and canine orthotopic xenograft murine models. For limb function evaluation, we evaluated lameness and grip capacity, and to assess ablation outcomes, we utilized ultrasound and histology for acute and chronic evaluation, respectively. Furthermore, the utility of magnetic resonance imaging (MRI) for characterizing orthotopic OS tumors and histotripsy ablation was investigated, and we evaluated the utility of the generated models for studying pulmonary metastatic disease burden. The current study is the first published investigation for the use of an orthotopic xenograft murine model for testing histotripsy ablation for OS; thus, the developmental process of the model, technical limitations of the utilized preclinical devices, and future directions will be addressed.

## 2. Materials and Methods

### 2.1. Xenograft Mouse Model

Female, 6–8-week-old CB17.Cg-Prkdc^scid^Lyst^bg-J^/Crl (*Scid* beige) (Strain Code 250, Charles River, Chapel Hill, NC, USA) mice were used. To generate the orthotopic human and canine xenograft models, 6 × 10^5^ 143B cells [24], or 1 × 10^6^ D17 cells, respectively, were suspended in a 50:50 1xPBS:Matrigel (Corning, Corning, NY, USA) solution and injected in the right hind limb via para-tibial injection, which is a well-established tumor cell implantation method [11,25,26]. Microscopic histological evaluation by a board-certified pathologist (S.C.O.) confirmed that the tumors originated from the periosteum in all mice, as seen in previously reported investigations [11,26].

Both the 143B and D17 cell lines (ATCC, Manassas, VA, USA) were maintained in high glucose DMEM (ATCC, Manassas, VA, USA) supplemented with 10% heat-inactivated FBS (Gibco Thermo Fisher Scientific, Waltham, MA, USA) and 1% penicillin/streptomycin (Gibco Thermo Fisher Scientific). The experimental groups for the human (143B) and canine (D17) xenograft models consisted of (1) treated tumor group; tumor-bearing mice that received histotripsy therapy (*n* = 6), (2) untreated mice; tumor-bearing mice that did not receive histotripsy therapy (*n* = 6), and (3) naïve; healthy, non-tumor-bearing mice (*n* = 6). The well-being of the mice was assessed three times weekly via weight monitoring, overall behavioral and clinical (attitude, general body condition, and appearance) assessments, and limb function evaluation. Mice were euthanized when an endpoint criterion was met, which consisted of a lameness score of 4, grip capacity score of −3 (Table 1), body weight loss of greater than 20% of initial body weight or at the conclusion of the study (90 days post-tumor induction). Mice were euthanized via CO_2_ inhalation and cervical dislocation was used as the secondary method. All experiments were conducted under an approved Virginia Tech Institutional Animal Care and use Committee (protocol number 20-151) and in accordance with the NIH Guide for the Care and Use of Laboratory Animals.

### 2.2. Limb Function Assessment

The assessment consisted of lameness and grip capacity evaluation, and each parameter was evaluated and scored by the same investigator throughout the study (A.N.H.). The scoring system is outlined in Table 1. To conduct the lameness assessment, mice were placed individually into a clean bedded cage and observed for 5 min to evaluate gait. For grip capacity assessment, mice were placed on a metal grate cage top, and ability to grip the metal grate with the tumor-bearing limb (right) was assessed relative to the non-tumor-bearing hind limb (left). Limb function data are presented as group data, and there was a decline in the total number of mice for each group as mice met endpoint criteria and were sacrificed (Appendix A).

### 2.3. Histotripsy Treatment

The 1 MHz histotripsy system utilized in the current study to achieve partial tumor volume ablation has previously been described [22,23]. Prior to the delivery of histotripsy, mice were anesthetized via inhaled isoflurane and positioned on the animal treatment bed in a manner that allowed complete submersion of the tumor-bearing limb in degassed water (Figure 1). Through the histotripsy procedure, mice were maintained under anesthesia. Histotripsy was delivered to an ellipsoidal volume within the tumor using a robotically guided automated treatment software that covered a predetermined region of the tumor [22,23]. Histotripsy was applied using single-cycle pulses delivered at a pulse repetition frequency (PRF) of 500 Hz, with treatment points spaced 1 mm in the axial direction and 0.5 mm in the other two dimensions to ensure overlap between bubble clouds at adjacent points. A 2 s dwell time at each treatment point was used for delivery of 1000 pulses/point. Ablation volume was selected based on the criteria of obtaining the largest tumor ablation possible within the restrictions of IACUC-approved anesthetic event duration (40 min) and sparing skin overlying the tumor to maintain animal well-being. The ablation volumes for all mice were measured based on ultrasound, utilizing the imaging probe and parameters previously described [20,22,23].

### 2.4. Tissue Collection and Histology

All tumors from both the human and canine xenograft groups were collected post-euthanasia and fixed in 10% buffered formalin (Azer Scientific, Morgantown, PA, USA) for histological evaluation. Additionally, lung tissue was collected and fixed in 10% buffered formalin for evaluation of pulmonary metastases.

All harvested formalin-fixed tumor samples collected were paraffin-embedded, sectioned in 5 µm sections, and stained with hematoxylin and eosin (H&E) for histological analysis. Tumor sections were blindly evaluated by a board-certified veterinary pathologist (S.C.O.) and assessed qualitatively and semi-quantitatively. Semi- quantitative assessments included review of a single, 5 µm H&E section for tumor necrosis with a scoring system. Tumor necrosis was defined as the presence of dead or dying cells with varying amounts of eosinophilic debris and clumped basophilic nuclear debris with a variably intact stroma. A score of 0–4 was assigned to each sample according to the following parameters: 0 = 0% of the tissue exhibits characteristic features of tumor necrosis, 1 = 1–25% of the tissue exhibits characteristic features, 2 = 26–50% of the tissue exhibits characteristic features, 3 = 51–75% of the tissue exhibits characteristic features, and 4 = 76–100% of the tissue exhibits characteristic features. Lung metastasis was manually counted as the total number of individual metastatic foci in a single 5 μm section of a harvested lung tissue as representative section of all lung lobes.

### 2.5. Fecal Corticosterone

As an assessment of stress, fecal samples were collected from all mice (tumor-bearing and naïve) prior to tumor cell injection, immediately before histotripsy, 1-week post-histotripsy, and at the time of sacrifice. For fecal sample collection, mice were moved to a clean cage, and fresh fecal samples were collected with clean forceps, transferred to microcentrifuge tubes, and stored at −20 °C until analysis. Fecal corticosterone was assessed via the Cayman Corticosterone ELISA kit (Cayman Chemical, Ann Arbor, MI, USA) following the manufacturer's recommended protocol. A 50 mg fecal sample was used for each mouse at each assessment timepoint. For each timepoint and group, *n* = 6.

### 2.6. Magnetic Resonance Imaging (MRI)

We preliminarily investigated the use of MRI for tumor monitoring for two reasons; (1) to overcome the challenge of grossly delineating the boundaries of orthotopic OS tumor, which is required for measuring tumors manually with calipers and (2) to continue to advance our goal of clinical advancement of histotripsy ablation for OS by enabling us to identify and monitor histotripsy ablation zones in tumors. Given that canine OS is the primary focus of our current clinical work, we selected the canine xenograft group for the preliminary MRI experiments to monitor tumor growth and evaluate histotripsy ablation (*n* = 3 mice). We utilized T2-weighted scans that were completed with a 9.4 Tesla high-field BioSpec 94/20 USR scanner(Bruker, Billerica, MA, USA). The MRI scans were acquired 24 h post-histotripsy ablation. MRIs were evaluated quantitatively using volumetric segmentations performed in the open-source software 3D Slicer (V5.8.1, www.slicer.org 17 July 2025). Untreated tumor (tumor) and treated tumor (ablation) were segmented using a grow-from-seeds approach, and volume and signal intensity were reported for all segmentations. Signal intensity was assessed post-segmentation.

### 2.7. Statistical Analysis

For group comparisons, across timepoints were analyzed with parametric two-way ANOVA or non-parametric mixed effect models with Tukey test for post-hoc comparisons, and pairwise comparisons were made with non-parametric Mann-Whitney Test. For analysis of survival outcomes, a mantel-cox test was used. MRI Signal intensity pre- to post-treatment was analyzed using paired *t*-tests. Shapiro-Wilk test was used to determine normality. Statistical analyses were computed with Graph Pad Prism Software (version 10). Statistically significant cut-off value was *p* < 0.05.

## 3. Results

### 3.1. Human Xenograft Model

#### 3.1.1. Ablative and Histological Outcomes; Histotripsy Results in Tumor Ablation

Histotripsy was delivered 18 days post-tumor cell injection, and an average tumor volume of 183 mm^3^ (±167 mm^3^) was targeted for a partial ablation of the tumor. Tumor ablation was evident on ultrasound by visualization of the acoustic cavitation bubble cloud and a hypoechoic region (ablation zone) post-histotripsy (Figure 2A–C). Tumor tissue damage was presumably evident by gross edema and bruising in comparison to pre-histotripsy (Figure 2D,E), which notably resolved by 3 days post-histotripsy (DPH) (Figure 2F), further supporting that the edema and bruising were a result of histotripsy-induced tumor tissue damage. Microscopically, histotripsy ablated tumor regions can be characterized by tumor necrosis acutely, but at the chronic sacrifice timepoint of this study, we did not observe statistically significant differences between untreated and treated tumors on histology, which was not unexpected (Figure 3A,B). We did not observe any statistically significant differences in metastatic disease burden between untreated tumor-bearing mice and those that received histotripsy ablation (Figure 3C,D).

#### 3.1.2. Clinical and Tumor-Bearing Limb Functionality Outcomes; Histotripsy Alters Tumor-Bearing Limb Function

Partial histotripsy ablation significantly (*p* = 0.002) extended survival compared to mice that did not receive histotripsy ablation of the primary tumor (Figure 4A). The histotripsy group had a median survival 5 days greater than untreated mice. Clinical lameness was initially observed at 7 days post-tumor cell injection (DPTC) in one mouse and was observed in all tumor-bearing mice (untreated and histotripsy) by 13 DPTC. Histotripsy ablation initially resulted in an increase in lameness, but within 3 DPH, lameness improved and was less than what was observed in untreated mice (Figure 4B). A decline in grip capacity was initially observed at 7 DPTC and declined in all tumor-bearing mice by 14 DPTC (Figure 4C). Histotripsy ablation improved grip capacity compared to untreated mice (Figure 4C). Limb function data are presented as group data, and there was a decline in the total number of mice for each group (histotripsy ablated and untreated) as mice met endpoint criteria and were sacrificed (see Appendix A for further details).

Fecal corticosterone levels were assessed prior to tumor inoculation and the delivery of histotripsy, 1-week post-histotripsy, and at the time of sacrifice. Histotripsy ablation resulted in a decrease in corticosterone levels 1-week post-histotripsy compared to untreated mice at the same timepoint and compared to pre-histotripsy concentrations. However, this decrease was not sustained, and by endpoint, corticosterone concentrations in the histotripsy group rose above baseline (pre-tumor cell injection) concentrations, and endpoint concentrations in histotripsy group were greater than untreated mice at endpoint. As expected, with no-tumor control mice, fecal corticosterone levels remained relatively unchanged throughout the study (Figure 4D).

### 3.2. Canine Xenograft Model

#### 3.2.1. Ablative and Histological Outcomes; Histotripsy Results in Tumor Ablation

Histotripsy was delivered on average at 42 DPTC, and an average tumor volume of 76 mm^3^ (±46 mm^3^) was targeted for ablation. Tumor ablation was evident on ultrasound by hypoechoic regions (Figure 5A–C). Prior to histotripsy ablation, we did not observe bruising and/or edema in the tumors (Figure 5D). Immediately post-histotripsy, we observed edema and bruising at the site of ablation, suggesting tumor tissue damage (Figure 5E). The bruising and edema resolved within 3 DPH, further suggesting that bruising and edema were a result of histotripsy (Figure 5F). Although we did observe slightly more tumor necrosis in treated tumors compared to untreated tumors, this was not statistically significant (Figure 6A,B). Similar to the human xenograft model, this was not unexpected due to the chronic sacrifice timepoint. We did not observe pulmonary metastatic disease development in the canine xenograft model (Figure 6C,D).

#### 3.2.2. Clinical and Tumor-Bearing Limb Functionality Outcomes; Histotripsy Alters Tumor-Bearing Limb Function

Histotripsy ablation significantly (*p* = 0.004) extended survival in mice that received histotripsy ablation of primary tumor compared to untreated mice (Figure 7A). The median survival of the mice in the histotripsy ablation group was 7 days greater than the untreated group. Of the six mice in the histotripsy group, one mouse did not meet endpoint criteria and was sacrificed at the end of the study duration of 90 days post-tumor cell injection (Figure 7A). Histotripsy ablation induced acute lameness compared to pre-histotripsy and compared to the untreated mice, but acute lameness improved 5–7 days post-histotripsy ablation. Histotripsy mice continued to demonstrate mild gait abnormalities throughout the remainder of the study, but these were not statistically significant when compared to untreated mice (Figure 7B). The progression of disease seemed to also influence lameness because untreated mice demonstrated the greatest level of lameness prior to the time of sacrifice, but this sharp increase in lameness was not observed in treated mice prior to sacrifice (Figure 7B). Grip capacity began to decline prior to histotripsy ablation, and ablation resulted in a transient greater decline in grip capacity compared to pre-histotripsy and the untreated mice (Figure 7C). However, by the end of the study, the untreated mice demonstrated a greater decline in grip capacity compared to those that received histotripsy ablation (Figure 7C). Limb function data are presented as group data, and there was a decline in the total number of mice for each group (histotripsy ablated and untreated) as mice met endpoint criteria and were sacrificed (see Appendix A for further details).

Fecal corticosterone levels were assessed prior to tumor inoculation, before the delivery of histotripsy, 1-week post-histotripsy, and immediately prior to sacrifice in all study groups (no-tumor control, untreated, and histotripsy). There were no statistically significant differences in fecal corticosterone concentrations between the untreated and histotripsy groups. There was a significant (*p* = 0.03) decrease in fecal corticosterone levels from baseline to the other assessed timepoints in the no-tumor control group, and this trend was also observed in the tumor-bearing mice, although not statistically significant (Figure 7D).

#### 3.2.3. Investigation of Primary Tumor Monitoring with Magnetic Resonance Imaging; Histotripsy Ablation Is Evident on MRI

The development of a preclinical model for studying histotripsy ablation outcomes is essential for clinical translation of histotripsy. Given our team’s current involvement in the development of histotripsy ablation as a treatment modality for canine OS, we selected the canine xenograft model in this study for MRI investigation. On T2-weighted MRI scans, tumor growth and histotripsy ablation were both evident (*n* = 3 mice). The tumor was observed as a hyperintense region on the MRI scan prior to histotripsy, and 24 h post-histotripsy, edema was evident by the hyperintense rim surrounding the tumor and peritumoral muscle (Figure 8A,B). The ablated tumor is hypointense compared to pre-histotripsy ablation, suggestive of intratumoral hemorrhage (Figure 8A,B). Additionally, MRI signal intensity was quantified. There is a trend towards decreased signal intensity on T2 MRI from pre- to post-treatment (15.7% decrease), suggesting increased hemorrhage after ablation (Figure 8C).

## 4. Discussion

This study aimed to characterize and assess the utility of preclinical orthotopic xenograft murine models to advance histotripsy-based OS studies. Our team has demonstrated the feasibility of delivering histotripsy ablation to OS primary tumors in canines [16,17,27] and in a heterotopic syngeneic mouse model [22]. However, there continues to be numerous gaps in knowledge regarding histotripsy ablation related to chronic ablative and functional outcomes of the tumor-bearing limb post-histotripsy. We hypothesized that the use of a murine orthotopic OS human and canine xenograft model could be used as a preclinical model to help address these knowledge gaps.

We established that histotripsy ablation can successfully be delivered to a partial volume of an orthotopic OS tumor in both human and canine OS xenograft murine models. Given the chronic evaluation goals of this study, we did not expect to observe obvious and extensive tissue ablation histologically due to tissue resorption that would likely start to occur post-histotripsy ablation [28]. However, we did observe hypoechoic regions on ultrasound immediately post-histotripsy ablation (Figure 2 and Figure 5). Our previous studies have shown that hypoechoic regions of tumor observed on ultrasound during and immediately after the delivery of histotripsy correlate well with areas of acute tumor necrosis and destruction histologically [22]. This indicates that there is a high likelihood that we achieved a degree of histotripsy ablation that would have been histologically evident acutely but was not at the chronic timepoints assessed in the current study, likely due to tissue resorption. Additionally, we observed tissue destruction and radiographic evidence of blood products (i.e., hemorrhage) post-histotripsy ablation on T2-weighted MRI scans (Figure 8B), further supporting that we achieved effective ablation. Future studies that do not aim to investigate survival and/or other chronic outcomes post-histotripsy ablation should include acute sacrifice timepoints for histological evaluation of histotripsy ablation in the xenograft models used in the current study. The addition of an acute timepoint to a future ablation efficacy-focused study will allow for a comprehensive histological evaluation of the targeted tumor for assessment of cell destruction.

There was a temporal increase in lameness and decrease in grip capacity post-histotripsy ablation. We hypothesize that the transient negative impact on limb function was a result of histotripsy-induced edema and bruising. As the histotripsy-induced edema and bruising resolved, we observed improvement in limb function. In the human xenograft model, after ~3 days post-histotripsy, results indicate that histotripsy resulted in improvement of lameness and grip capacity for the human xenograft model (Figure 4B,C). For the canine xenograft model, grip capacity improved in the histotripsy ablated tumor group ~7 days post-histotripsy (Figure 7C). These results suggest that histotripsy may have positive impacts on tumor-induced limb dysfunction. We hypothesize that the observed changes in lameness and grip capacity were either a result of debulking the tumor (improving limb mobility), histotripsy-induced analgesic effect, or a combination of both. While we observed an improvement in tumor-bearing limb function post-histotripsy ablation compared to the untreated tumor-bearing mice, we did not completely alleviate tumor-induced limb dysfunction because mice continued to demonstrate some degree of limb dysfunction in both the human and canine xenograft groups. We hypothesize that this was likely due to residual non-ablated tumor in the tumor region, which was not targeted with histotripsy, resulting in continued limb dysfunction. Furthermore, our results suggest that histotripsy ablation improved the overall disease outcomes of the tumor-bearing mice by extending survival (Figure 4A and Figure 7A). In further support of histotripsy ablation extending survival, one of the canine xenograft model mice that received histotripsy ablation did not meet the defined endpoint criteria and was sacrificed at 90 days post-tumor cell injection (the end of the study duration). Thus, survival in the canine xenograft model is extended relative to the untreated tumor-bearing mice, but the true timepoint at which all the mice that received histotripsy ablation would have met study endpoint criteria was not determined in this study.

Additionally, in the human xenograft model, we observed a decrease in fecal corticosterone concentrations at 1-week post-histotripsy ablation, but this was not sustained over time, potentially due to continued tumor growth post-histotripsy ablation or because the potential stress-reducing effects of histotripsy ablation were transient (Figure 4D). Previous preclinical rodent model studies have reported increased fecal corticosterone levels in parallel with stress and discomfort [29,30], suggesting that histotripsy may have reduced stress and discomfort in the human xenograft model. In the canine xenograft model, we observed a decline in fecal corticosterone concentrations throughout the study compared to baseline (pre-tumor cell injection) across all study groups, suggesting that an external environmental factor, such as acclimation to human interaction, influenced the observed changes (Figure 7D). Although the current study was designed to assess functional and ablative outcomes post-histotripsy and not palliative outcomes, these results suggest that histotripsy ablation may improve tumor-associated discomfort. This aligns with previous investigations reporting that thermal high-intensity focused ultrasound partially or completely alleviated pain in human bone cancer patients [31,32,33,34]. Additionally, in human OS patients who receive thermal FUS treatment, a transient improvement in pain in response to treatment has previously been reported [34] similar to the transient responses we observed in this murine xenograft model study. In the current study, improvement of tumor-associated discomfort was especially notable in the human xenograft model, which had larger tumors compared to the canine xenograft model, suggesting that tumor size may impact ablative outcomes. Future investigation into the use of histotripsy for palliative care, with careful consideration of tumor size, is warranted.

Our results indicate that MRI will also have high utility for monitoring OS tumor growth and progression post-ablation (Figure 8). The use of MRI will address the challenge of gross delineation of orthotopic OS tumor boundaries, which is needed for manual caliper measurements and allows for more accurate measurements via imaging analysis software such as 3D Slicer. If we have the ability to accurately measure tumor volumes, histotripsy can be delivered when tumor growth reaches a specific volume, and tumor progression post-ablation can be monitored over time in future studies. Additionally, methods such as measuring MRI signal intensity can be utilized for ablation assessment. This has a high degree of relevance to clinical application of histotripsy ablation, given that MRI imaging is a commonly used modality in both human and veterinary medicine for monitoring bone tumors, and non-invasive methods for monitoring ablation will be valuable [35,36,37].

The overall goal of this study was to investigate both human and canine orthotopic xenograft murine models for histotripsy ablation of OS and independently evaluate each model. We did not have the intention of directly comparing the outcomes between the two models. However, it is worth acknowledging that we did observe differences in tumor growth rate; tumors composed of human-143 B cells progressed at a more rapid rate than canine-D17 cell tumors. Since the goal of this study was to distinctly assess outcomes relative to histotripsy ablation in each model, the interspecies tumor growth differences did not negatively affect our study goals. Furthermore, lung metastasis was observed in the human xenograft model but not in the canine model. The metastatic potential of the 143B cell line in an orthotopic murine model has previously been reported [38,39]. The D17 cell line has previously been reported to be utilized for a canine OS xenograft murine model [40], which also did not report lung metastasis. However, the use of other canine OS cell lines, such as the Abrams cell line, in xenograft models has shown evidence of pulmonary metastasis [41,42]. These previous reports indicate a low level of pulmonary metastases present at 3 months post-orthotopic tumor induction [41], suggesting that the development of metastatic disease in our orthotopic canine xenograft model may require more than a 3-month (90-day) timeframe to develop. Our observations of metastatic disease progression suggest that the 143B cell line is suitable for future histotripsy-based studies, which aim to investigate metastatic disease using an orthotopic OS xenograft model, but the canine xenograft model will require further development for the purpose of metastatic disease investigations.

Due to the pilot nature of this study, a limitation was that we had to deliver histotripsy to a volume of the tumor that could be ablated within the permitted anesthetic event duration of 40 min (including set-up time), and while we aimed to target the largest volume possible within the tumors, a ~1 mm region of tumor closest to the skin was left intact to prevent excessive pre-focal cavitation-induced damage to the skin overlying the tumor. Pre-focal ablation of the overlying skin would have resulted in greater tumor ablation volumes but would have negatively impacted the well-being of the mice, and, subsequently, our downstream assessments of limb function and stress. Future studies with histotripsy devices that have the capability to ablate closer to the skin or investigation of doses to spare skin overlying the tumors would be beneficial. Future studies should also consider tumor heterogeneity and the potential of bone-blockage of acoustic cavitation, and the effect this may have on cavitation consistency.

Perhaps the most exciting potential of the murine xenograft model is the potential it has to aid in the advancement of precision medicine [43,44]. While the use of syngeneic orthotopic mouse models [26,45,46] has routinely been utilized for investigation of OS, the translational utility of these investigations to human and/or veterinary medicine is limited [11]. As previously mentioned, our team has successfully ablated canine OS both ex vivo and in vivo, exemplifying the well-established utility of the canine patient in the development of novel OS treatment options. However, as our team continues to advance histotripsy ablation as a potential treatment option for both canine and human OS, a preclinical model that allows us to capture species-specific oncological and ablative responses to histotripsy ablation will be incredibly beneficial. One of the major challenges for developing treatment options for OS is intra- and inter-patient tumor heterogeneity, and the use of a PDX mouse model addresses this challenge. Although the current study did not use a PDX model, the species-specific cell line xenograft model utilized in the current study lays the foundation for future studies aiming to investigate species-specific responses to tumor ablation and paves the way for characterization of patient-specific responses with PDX models. Additionally, future studies will be conducted to combine histotripsy with immunotherapies, such as the immune check inhibitors anti-CTLA4 and anti- PD1. We hypothesize that the combination of immunotherapies with histotripsy will result in a synergistic effect and improve the efficacy of both histotripsy and the immunotherapy, resulting in more robust metastatic disease mitigation. In conclusion, the results of this pilot study establish the possibility of the use of human and canine orthotopic xenograft murine models to advance the novel tumor ablation modality histotripsy for OS by investigating functional and ablative outcomes.

## Figures and Tables

**Figure 1 biomedicines-13-02122-f001:**
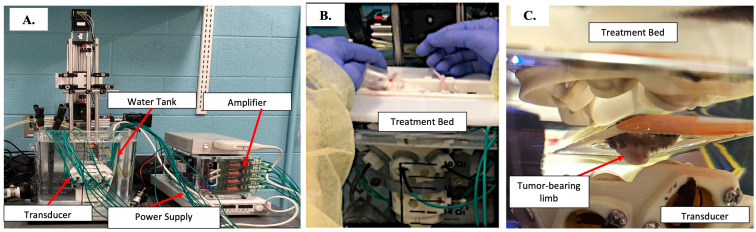
Images of histotripsy system and setup: (**A**) Image depicts the histotripsy system with essential pieces of equipment labeled. (**B**) Image depicts a mouse lying in the treatment bed positioned above the transducer. (**C**) The right hind limb bearing an OS tumor is submerged into the water tank for histotripsy ablation.

**Figure 2 biomedicines-13-02122-f002:**
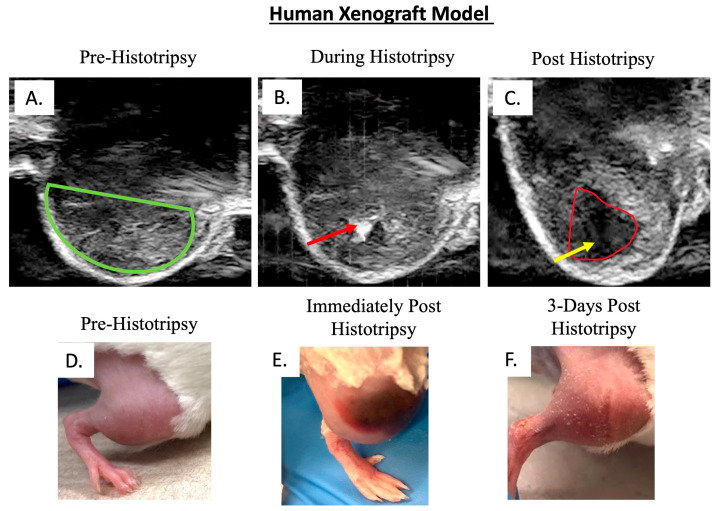
Representative images of human xenograft mouse model tumors. B-mode ultrasound images of (**A**) tumor before histotripsy (green outline), (**B**) histotripsy bubble cloud (red arrow), and (**C**) hypoechoic region of tumor targeted by histotripsy, ablation zone (red outline and yellow arrow). (**D**) Image of tumor-bearing limb prior to histotripsy, (**E**) image depicting the bruising and edema post histotripsy, and (**F**) 3 days post-histotripsy, bruising and edema are resolved.

**Figure 3 biomedicines-13-02122-f003:**
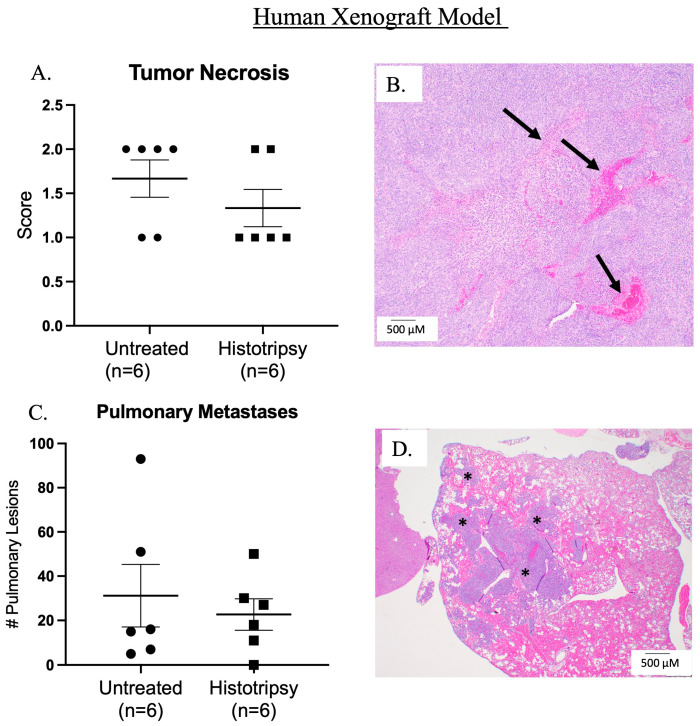
Histological assessment of histotripsy ablation in human xenograft model. (**A**) Scatter plot of tumor necrosis score in both untreated tumors and histotripsy ablated tumors. (**B**) Representative image of tumor necrosis in a histotripsy ablated tumor. Arrows indicate areas of tumor necrosis and hemorrhage. (**C**) Scatter plot of total pulmonary metastases in histologically assessed lung tissue collected from tumor-bearing mice. (**D**) Representative image of pulmonary metastases, asterisks represent examples of lung metastases. Error bars represented as SEM.

**Figure 4 biomedicines-13-02122-f004:**
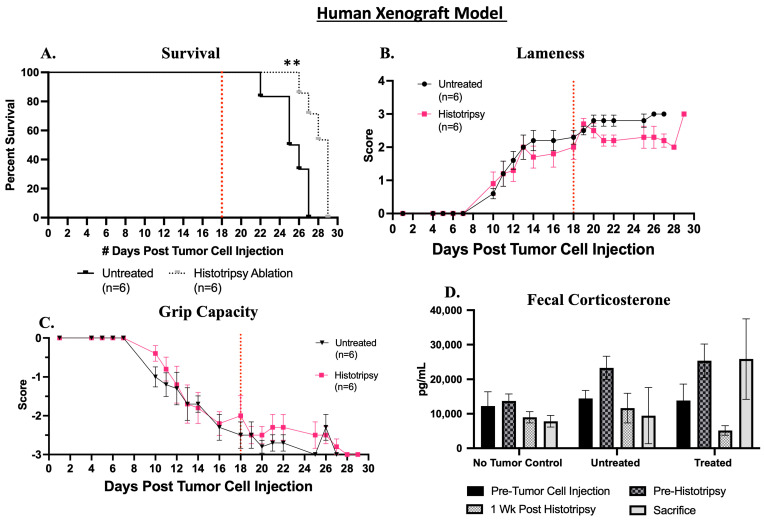
Assessment of survival and functional outcomes post-histotripsy in human xenograft model. (**A**) Human xenograft model; we observed that histotripsy ablation resulted in a significantly greater (** *p* = 0.002) survival. (**B**) Lameness outcomes, (**C**) Grip capacity outcomes, and (**D**) Fecal corticosterone concentrations. The mouse groups are no-tumor control (naïve mice), untreated (tumor-bearing untreated mice), and histotripsy (tumor-bearing mice that received histotripsy ablation). For each timepoint and group, *n* = 6. Limb function data (lameness and grip capacity) are presented as group data, and there was a decline in the total number of mice for each group (untreated and histotripsy) as mice met endpoint criteria and were sacrificed. Further details can be found in Appendix A. Red dashed indicates average day of histotripsy ablation. Error bars represent SEM.

**Figure 5 biomedicines-13-02122-f005:**
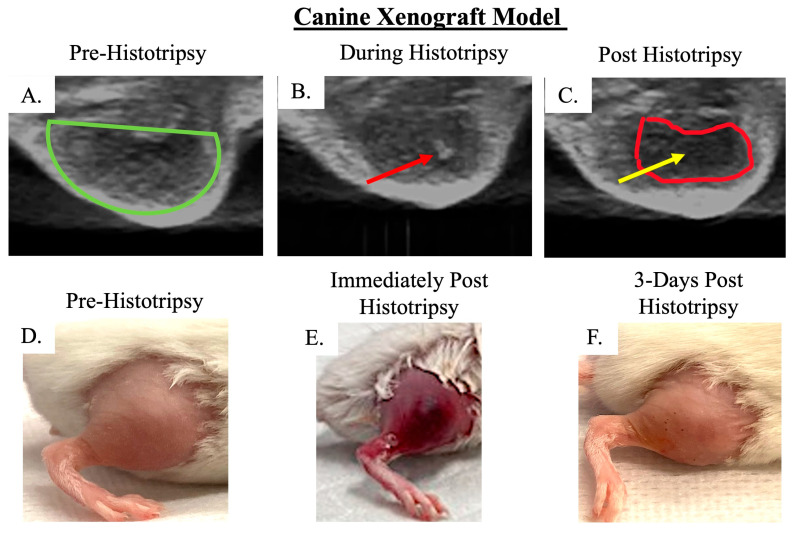
Representative images of canine xenograft mouse model tumors. B-mode ultrasound images of (**A**) tumor before histotripsy (green outline), (**B**) histotripsy bubble cloud (red arrow), and (**C**) hypoechoic region of tumor ablated by histotripsy (red outline and yellow arrow). (**D**) Image of tumor-bearing limb prior to histotripsy, (**E**) image depicting the bruising and hemorrhage post-histotripsy, and (**F**) 3 days post-histotripsy, hemorrhage and bruising are resolved.

**Figure 6 biomedicines-13-02122-f006:**
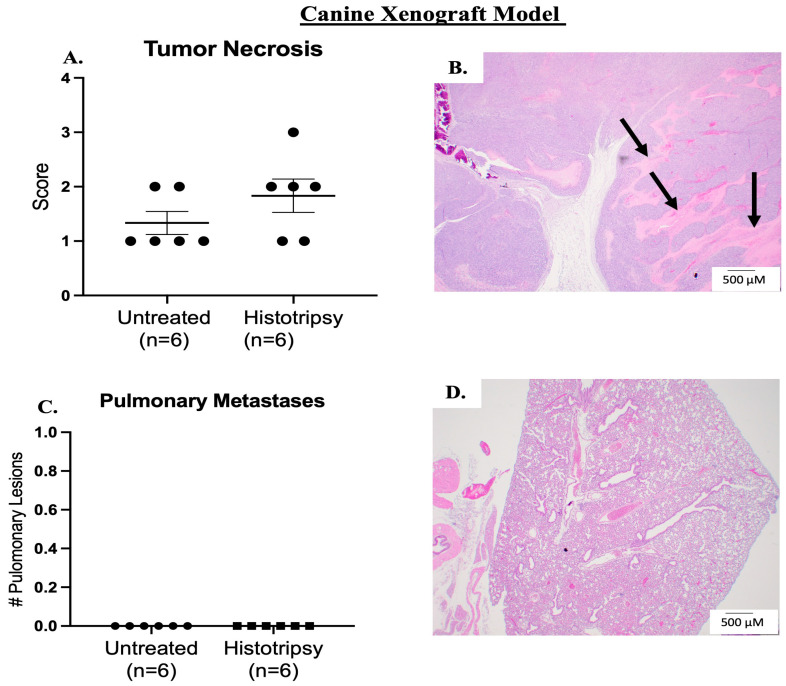
Histological assessment of histotripsy ablation in canine xenograft model. (**A**) Scatter plot of tumor necrosis score in both untreated tumors and histotripsy ablated tumors. (**B**) Representative image of tumor necrosis in a histotripsy ablated tumor, arrows indicate areas of tumor necrosis. (**C**) Scatter plot of total pulmonary metastases in histologically assessed lung tissue collected from tumor-bearing mice. We observed no metastases in untreated or histotripsy-ablated tumor-bearing mice. (**D**) Representative image of pulmonary metastases. Error bars represented as SEM.

**Figure 7 biomedicines-13-02122-f007:**
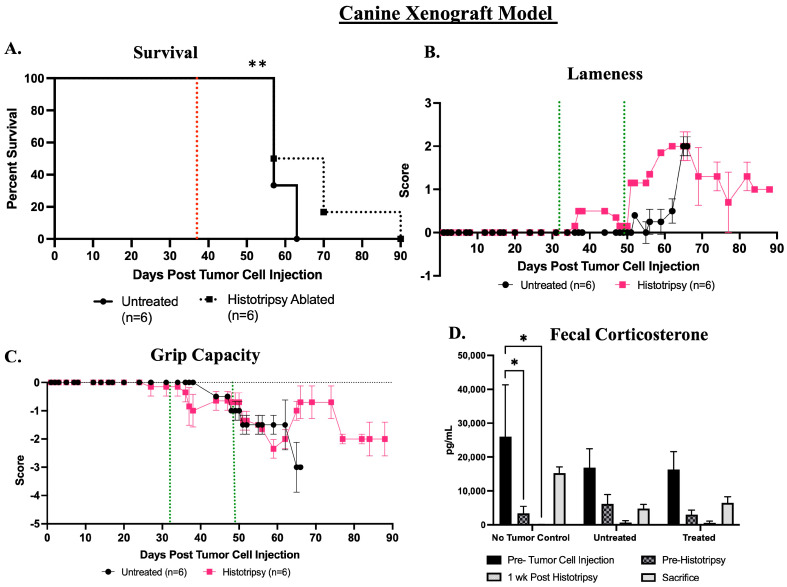
Assessment of survival and functional outcomes post-histotripsy in canine model. (**A**) histotripsy ablation resulted in a significantly extended (** *p* = 0.004) survival. Red dashed line represents average day post-tumor cell injection when histotripsy was delivered to both canine model groups. (**B**) Lameness outcomes and (**C**) Grip capacity outcomes. Green dashed line indicates average day of delivery of histotripsy ablation for each canine model group. (**D**) The fecal corticosterone concentrations (* *p* = 0.03). The mouse groups are no-tumor control (naïve mice), untreated (tumor-bearing untreated mice), and histotripsy (tumor-bearing mice that received histotripsy ablation). For each timepoint and group, *n* = 6. For limb function, data (lameness and grip capacity) are presented as group data, and there was a decline in the total number of mice for each group (histotripsy ablated and untreated) as mice met endpoint criteria and were sacrificed (see Appendix A for further details). Error bars represent SEM.

**Figure 8 biomedicines-13-02122-f008:**
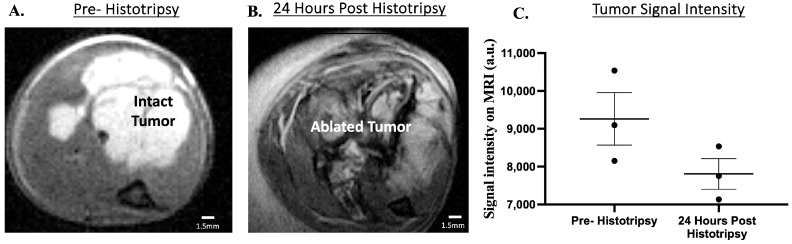
Representative MRI images of canine xenograft tumors and quantitative assessment of signal intensity. (**A**,**B**) Images were acquired with a T2-weighted sequence. The tumor is the hyperintense region in the pre-histotripsy image. At 24 h post-histotripsy, tumor destruction and edema are evident. (**C**) T2-MRI signal intensity pre- and post-histotripsy ablation. Error bars represent SEM.

**Table 1 biomedicines-13-02122-t001:** Lameness and grip capacity scoring chart. * Reduction is relative to non-tumor-bearing hind limb.

Lameness		Grip Capacity *	
0	No lameness	0	Normal grip
1	Slight gait abnormality.	−1	Mild reduction
2	Full range of motion but moderate gait abnormality (ex. Limping). Sensation in toes present.	−2	Severe reduction
3	Impaired range of motion. Sensation in toes present.	−3	No Grip capacity (Endpoint)
4	Not weight bearing, absence of sensation in toes (End point).		

## Data Availability

The original contributions presented in this study are included in the article/Appendix A. Further inquiries can be directed to the corresponding author.

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
