# Peer review of "Establishing Human and Canine Xenograft Murine Osteosarcoma Models for Application of Focused Ultrasound Ablation"

_biomedicines, 2025, doi:10.3390/biomedicines13092122_

Round 1
Reviewer 1 Report (Previous Reviewer 2)
Comments and Suggestions for Authors
The authors have provided satisfactory responses to the reviewers comments.
Author Response
Thank you for the feedback and approval.
Reviewer 2 Report (New Reviewer)
Comments and Suggestions for Authors
In this article, Hay et al aimed to establish mouse models of human and canine osteosarcoma for the purpose of testing histotripsy applications, orthotopically. The authors showed the successful engraftment of the tumors, treatment with histotripsy and assessment of the tumors clinically and histologically. This is an important pilot study, paving the way for further characterization and personalized treatment for humans and dogs. Here are some suggestions for the authors:
Major comments
1) All Tables and Figures are low resolution images. Please replace with high resolution equivalents.
2) Figure 4 shows that animals started dying after Day 22 until there was one animal left at Days 26 and 28 for untreated and treated mice respectively. This is well documented at Fig4A as statistically significant. However, these time points cannot be adequately compared for the rest of the graphs, since animal groups are not complete. Please make sure that the only comparisons in lameness, grip capacity and fecal corticosterone are presented only until the last time point where all the animals were present. Same rationale applies to Figure 7 for the canine model.
3) For the time points with all animals present in the groups, were any of the differences shown at Fig4B, C or D statistically significant? Please adjust the narrative in the results and discussion sections to emphasize true, statically significant changes and distinguish them from visual trends. This is applicable to all figures and comparisons discussed.
4) It is important to briefly emphasize the important of the dog as naturally occurring model of human cancer.
5) There are many sentences throughout the manuscript that are highlighted. Please review them.
Minor comments
6) Line 6: Please fix the gap.
7) Line 63: Please add "the murine xenograft model"
8) Line 77: "heterotypic". Do the authors mean heterotopic?
9) Line 106: "50% 1xPBS:Matrigel". Do the authors mean 50:50 1xPBS:Matrigel?
10) Line 123: Did the authors used any secondary method for euthanasia?
11) Table 1 is still not editable and the table number is shown with latin numerals
12) Line 205: How did the authors evaluate the assumptions of the parametric tests used?
13) Lines 211, 239-240, Line 268 and 293-294: Please format the subheading.
14)Lines 224-225. Since the difference was not statistically significant, please remove this statement.
15) Figures 1, 2, 3, 4, 5, 6, 7, 8: Please remove the figure legend that is embedded in the photo.
16) Line 238 and 292: Do the authors mean that the data are shown as mean +- SEM? Please also add that sentence to the Figure legend of Figure 8.
17) Lines 373-373: Please remove "to an"
Author Response
Please see attached response letter.

Round 2
Reviewer 2 Report (New Reviewer)
Comments and Suggestions for Authors
I would like to thank the authors for addressing most of my comments.
Regarding comment #2, I agree that everything should be reported and either create individual graphs for each animal/group or clearly indicate the number of animals left per time point in the current graphs. However, statistically significant comparisons cannot be made between the groups. Qualitative description of the data is sufficient.
Author Response
Please see attached letter.

This manuscript is a resubmission of an earlier submission. The following is a list of the peer review reports and author responses from that submission.
Round 1
Reviewer 1 Report
Comments and Suggestions for Authors
The authors present a preclinical study evaluating the efficacy of histotripsy in an orthotopic osteosarcoma murine model using both human (143B) and canine (OS-49) OS cell lines. This study builds upon their previous work in subcutaneous models and aims to enhance clinical relevance by testing tumor ablation in an intratibial (orthotopic) setting. While the orthotopic approach adds novelty, several aspects of the study require clarification and further strengthening. Histotripsy is a non-invasive, non-thermal ultrasound-based ablation technique that mechanically destroys tissue through focused ultrasound pulses. Unlike traditional thermal ablation methods (e.g., radiofrequency or microwave), histotripsy relies on acoustic cavitation to disrupt tissue. Bone's high acoustic impedance makes it difficult for ultrasound to penetrate, which may explain why the orthotopic model in this study showed no tumor cell death. Nevertheless, partial effects such as sublethal mechanical disruption or microenvironmental modulation could yield anti-inflammatory or immunomodulatory responses, rather than tumor necrosis.
Major Comments:
Introduction
The statement: "Canine OS occurs at a greater prevalence than human OS" should be supported with a reference to epidemiological data.
Methods
- The rationale for selecting 6 × 10⁵ 143B cells for implantation is unclear. Additional justification is needed, possibly citing prior studies such as Martella et al. used 10 × 10⁶ Saos-2 cells in an orthotopic murine model to evaluate photodynamic therapy (see Martella et al., 10.3390/pharmaceutics14030677).
- The authors should either cite relevant references or explain their choice of cell line and seeding density, especially in light of interspecies and tumor biology differences.
- Although the authors claim in the discussion: "Since the goal of this study was to distinctly assess outcomes relative to histotripsy ablation in each model, the interspecies tumor growth differences did not negatively affect our study goals," it is unclear if this was the original goal. It appears that direct comparisons between tumor types were made, and this may undermine some conclusions unless better justified.
- The number of animals per group, and the breakdown of those receiving human vs. canine cells, should be explicitly stated in the Methods section.
- Greater transparency is needed in describing model development, including:
- Tumor implantation technique
- Tumor take rates
- Differences in tumor invasiveness, metastatic potential, or growth kinetics between the cell lines
- The image quality in Figures 3 and 6 is too low to reliably assess tumor necrosis. Consider re-uploading higher-resolution images.
- A TUNEL assay or similar histological staining should be considered to better quantify tumor necrosis.
Results
- The sentence: "Ablative and histological outcomes; histotripsy results in tumor ablation." appears to be a heading or incomplete. If it's a heading, format it accordingly; if not, revise to a complete sentence.
- Clarify how tumor volumes (e.g., 183 mm³ ± 167 mm³) were measured. Was this via ultrasound and if yes how was performed and in how many animals? Imaging modality, frequency, and resolution should be described. Figures should also include scale bars.
- Was limb function assessed using a validated scoring system or objective behavioral metrics?
- The term "non-acute sacrifice timepoint" (lines 203, 261) needs explanation. What was the timing and purpose of this endpoint?
- Figure 8 lacks quantification. As it stands, the data presented are qualitative and offer limited value without supporting numerical analysis.
Discussion
- The authors correctly note that this is the first report using an orthotopic xenograft model to evaluate histotripsy in OS, which is a step forward from subcutaneous models. However, no statistically significant differences were observed between treated and control groups. This should be acknowledged more explicitly, along with a critical discussion of potential reasons:
- Inadequate acoustic energy reaching the tumor
- Limitations in targeting precision
- Tumor heterogeneity
- The discussion could benefit from more detail on technical limitations, such as:
- Cavitation consistency
- Ultrasound focusing in bone
- Variability in tumor anatomy
- It appears that the authors expected tumor ablation but did not observe it, possibly due to acoustic impedance. Instead of framing the paper primarily as a proof of concept for the mouse model, they could strengthen their findings by exploring alternative interpretations—for example, that histotripsy did not ablate the tumor but may have altered the tumor microenvironment.
- Incorporating histological analysis with inflammatory markers (e.g., CD45, IL-6, F4/80) could help determine whether immune or anti-inflammatory responses were triggered by sublethal treatment.
- The authors might consider conducting ex vivo experiments using decellularized or 3D-printed bone models to test different ultrasound protocols and optimize parameters before proceeding to in vivo studies. This would reduce animal use and allow better understanding of acoustic propagation in bone.
Minor Comments
- The manuscript contains some structural issues that affect readability and many repetitions of the same concepts. A thorough stylistical revision is recommended.
- A graphical abstract or summary figure would greatly enhance clarity and highlight the experimental design and outcomes.
Author Response
We thank the reviewer for their detailed feedback. Please see the attached document for author responses. We hope our responses and edits are satisfactory.
|
Response to Reviewer 1 Comments
|
|||
|
1. Summary |
|
|
|
|
Thank you very much for taking the time to review this manuscript. Please find the detailed responses below and the corresponding revisions/corrections highlighted and/or the corresponding line numbers are provided.
Sincerely, Alayna Hay, et al. |
|||
|
|
|
||
|
|
|
||
|
|
|
||
|
|
|
||
|
|
|
||
|
Point-by-point response to Comments and Suggestions for Authors Reviewer 1 |
|||
|
Comments 1: "Canine OS occurs at a greater prevalence than human OS" should be supported with a reference to epidemiological data. Response 1: Thank you, supporting references have been added.
Comment 2: The rationale for selecting 6 × 10⁵ 143B cells for implantation is unclear. Additional justification is needed, possibly citing prior studies such as Martella et al. used 10 × 10⁶ Saos-2 cells in an orthotopic murine model to evaluate photodynamic therapy (see Martella et al., 10.3390/pharmaceutics14030677). Response 2: Thank you for this suggestion. A relevant publication that informed our selection of total number of cells for the 143B tumor induction has been added to the manuscript in the materials and methods section.
Comment 3: The authors should either cite relevant references or explain their choice of cell line and seeding density, especially in light of interspecies and tumor biology differences. Response 3: Thank you for your comment. The selection of D17 and 143B cell lines were chosen because they are commonly used for preclinical canine and human OS research. The review brings up an excellent point about interspecies differences and consequently tumor biology differences. However, as discussed in the manuscript, despite the interspecies differences, canine and human OS share numerous similarities making results from both species informative. The seeding density utilized for tumor induction was based on previous literature which has been added to the manuscript.
Comment 4: Although the authors claim in the discussion: "Since the goal of this study was to distinctly assess outcomes relative to histotripsy ablation in each model, the interspecies tumor growth differences did not negatively affect our study goals," it is unclear if this was the original goal. It appears that direct comparisons between tumor types were made, and this may undermine some conclusions unless better justified. Response 4: Thank you for the feedback. The goal of the study was to evaluate the two models individually. The authors have read over the manuscript to improve clarity of this overall goal. If the reviewer has specific concerns regarding conclusions made in the manuscript, the authors are happy to consider them and address them.
Comment 5: The number of animals per group, and the breakdown of those receiving human vs. canine cells, should be explicitly stated in the Methods section. Response 5: Thank you for pointing out the need for further clarity. Additional content has been added to the methods section (lines 107-108, Xenograft mouse model section) for further clarity.
Comment 6: Greater transparency is needed in describing model development, including:
Response 6: Thank you for the feedback. The tumor implantation technique can be found in the materials and methods section Xenograft mouse model, first paragraph (~lines 97-103). The tumor engraftment rate was 100% and this was clarified in the methods section. Information regarding the differences in the tumor cell lines is discussed in the discussion lines. If the reviewer has additional specific requests for further relevant discussion the authors are happy to consider including additional content.
Comment 7: The image quality in Figures 3 and 6 is too low to reliably assess tumor necrosis. Consider re-uploading higher-resolution images. Response 7: Thank you for the feedback. Higher resolution images have been provided to MDPI for publishing.
Comment 8: A TUNEL assay or similar histological staining should be considered to better quantify tumor necrosis. Response 8: Thank you for the feedback. The reviewer’s suggestion is insightful, but outside the scope of this study. A board-certified veterinarian pathologist with extensive experience in reviewing tumor ablation reviewed the histology for this manuscript, and we are confident in our results. Results Comment 9: The sentence: "Ablative and histological outcomes; histotripsy results in tumor ablation." appears to be a heading or incomplete. If it's a heading, format it accordingly; if not, revise to a complete sentence. Response 9: Thank you for feedback. As suggested by the reviewer, this is a heading, and we have requested that the publisher reformats this accordingly.
Comment 10: Clarify how tumor volumes (e.g., 183 mm³ ± 167 mm³) were measured. Was this via ultrasound and if yes how was performed and in how many animals? Imaging modality, frequency, and resolution should be described. Figures should also include scale bars. Response 10: Thank you for the feedback. The ultrasound ablation volume was calculated by ultrasound and for all animals this information has been added to the materials and methods (lines 143-145), and a scale bar has been added to the figures.
Comment 11: Was limb function assessed using a validated scoring system or objective behavioral metrics? Response 11: Thank you for your question. The limb function assessment is described in section 2.2 limb function assessment in the materials and methods as an objective behavioral metric. The authors are unsure if the reviewer would like further details but are happy to provide further details if needed.
Comment 12: The term "non-acute sacrifice timepoint" (lines 203, 261) needs explanation. What was the timing and purpose of this endpoint? Response 12: Thank you for your question. The authors have changed “non-acute” to “chronic”, for improved clarity.
Comment 13: Figure 8 lacks quantification. As it stands, the data presented are qualitative and offer limited value without supporting numerical analysis. Response 13: Thank you for the comment. The authors are aware that the pilot data present in figure 8 is qualitative but feel that it is valuable data. The authors are unsure what change or question the reviewer is requesting/asking regarding figure 8. Discussion Comment 14: The authors correctly note that this is the first report using an orthotopic xenograft model to evaluate histotripsy in OS, which is a step forward from subcutaneous models. However, no statistically significant differences were observed between treated and control groups. This should be acknowledged more explicitly, along with a critical discussion of potential reasons:
Response 14: Thank you for the feedback and the reviewer brings up valid points to consider. The relevant limitations of the study are acknowledged and discussed in the discussion.
Comment 15: The discussion could benefit from more detail on technical limitations, such as:
Response 15: Thank you for this suggestion. The authors think that the reviewer brings up valid points and these points have been added to the discussion.
Comment 16: It appears that the authors expected tumor ablation but did not observe it, possibly due to acoustic impedance. Instead of framing the paper primarily as a proof of concept for the mouse model, they could strengthen their findings by exploring alternative interpretations—for example, that histotripsy did not ablate the tumor but may have altered the tumor microenvironment. Response 16: Thank you for your feedback. The authors do think ablation was observed via ultrasound and gross assessment, along with limb function improvement. However, tumor growth continued likely due to inadequate ablation volume, not ineffective ablation. The manuscript has been reviewed to ensure clarity of this point, and also additional limitations have been added to the discussion.
Comment 17: Incorporating histological analysis with inflammatory markers (e.g., CD45, IL-6, F4/80) could help determine whether immune or anti-inflammatory responses were triggered by sublethal treatment. Response 17: Thank you for this suggestion but the immunocompromised state of this mouse model does not allow for reliable assessment of the immune response associated with histotripsy ablation. Future studies involving immunocompetent mice will be conducted to evaluate the immune response.
Comment 18: The authors might consider conducting ex vivo experiments using decellularized or 3D-printed bone models to test different ultrasound protocols and optimize parameters before proceeding to in vivo studies. This would reduce animal use and allow better understanding of acoustic propagation in bone. Response 18: Thank you for this feedback and insightful suggestion. The authors will consider this for future studies. Minor Comments Comment 19: The manuscript contains some structural issues that affect readability and many repetitions of the same concepts. A thorough stylistical revision is recommended. Response 19: Thank you for the feedback. Can the reviewer please provide further detail on their request? The authors would like to point out that since two different mouse models were reported on (canine and human xenograft) there is a bit of a repetitive nature to the format of the paper but the content is not repetitive.
Comment 20: A graphical abstract or summary figure would greatly enhance clarity and highlight the experimental design and outcomes. Response 20: Thank you for this suggestion. At this time the authors have selected not to create a graphical abstract as we feel the figures in the manuscript provide a sufficient summary of the manuscript. However, if the editors and reviewers feel strongly about the need for a graphical abstract the authors will reconsider.
|
|||
Reviewer 2 Report
Comments and Suggestions for Authors
This is an interesting and well executed study of humane and canine OS xenografts ablation by focused ultrasound. In its current form this approach seems to be suitable for single tumors. However, in both human and canine patients as well as in the models used by the authors the metastatic spread to distant organs is of outmost importance. Therefore, the authors should include in the Discussion their thoughts on how the ultrasound treatment could be combined with systemic therapies pre-clinically and clinically to address the problem of metastases.
Author Response
Comment 1: This is an interesting and well executed study of humane and canine OS xenografts ablation by focused ultrasound. In its current form this approach seems to be suitable for single tumors. However, in both human and canine patients as well as in the models used by the authors the metastatic spread to distant organs is of outmost importance. Therefore, the authors should include in the Discussion their thoughts on how the ultrasound treatment could be combined with systemic therapies pre-clinically and clinically to address the problem of metastases.
Response 1: Thank you for the supportive feedback. We had added combination therapies to our discussion (lines 457-461).
Round 2
Reviewer 1 Report
Comments and Suggestions for Authors
Thank you for addressing several of the points I raised in my previous review. I appreciate the clarifications and revisions made. However, I still believe that the intended scope of using histotripsy to ablate the tumor has not been demonstrated therefore I'm not sure what value this study is adding in the field. In particular, I find the presentation of the MRI data to be insufficient; a quantitative analysis would add significant value, rather than relying on a single representative image per group. The authors’ response that “ablation was observed via ultrasound and gross assessment” is not sufficient to address my comment. That said, I will defer to the editor’s judgment regarding whether the manuscript in its current form merits publication.
Author Response
Reviewer Comment: Thank you for addressing several of the points I raised in my previous review. I appreciate the clarifications and revisions made. However, I still believe that the intended scope of using histotripsy to ablate the tumor has not been demonstrated therefore I'm not sure what value this study is adding in the field. In particular, I find the presentation of the MRI data to be insufficient; a quantitative analysis would add significant value, rather than relying on a single representative image per group. The authors’ response that “ablation was observed via ultrasound and gross assessment” is not sufficient to address my comment. That said, I will defer to the editor’s judgment regarding whether the manuscript in its current form merits publication.
Authors Response: Thank you for the feedback. The authors have added additional quantitative analysis for the presented MRI data. Please see revised figured 8 and the corresponding methods, results, and discussion sections (highlighted in yellow). Respectfully, the authors would also like to reiterate that the purpose of the manuscript was not to assess acute ablation efficacy but rather assess the chronic outcomes addressed in the manuscript. The authors acknowledge the importance of assessing ablation efficacy in future studies which will be conducted to further advance histotrispy ablation for OS and clinical application. The authors feel that the development of a preclinical rodent model will have high utility for advancing histotripsy and will also provide a feasible model for evaluation of chronic-long term ablative and oncological outcomes. We hope the additional revisions are statisfactory and greatly appreciate the efforts and time of the reviewer. We feel that the reviewer's feed back has greatly improved our manuscript, thank you.